# Quadrupole anomalous Hall effect in magnetically induced electron nematic state

Hiroki Koizumi ®[1,2,3] ✉, Yuichi Yamasaki ®[4] ✉ & Hideto Yanagihara ®[1,5] ✉

Berry phases in both momentum and real space cause transverse motion in itinerant electrons, manifesting various off-diagonal transport effect such anomalous and topological Hall effects. Although these Hall effects are isotropic within the plane perpendicular to the fictitious magnetic field, here, we report the manifestation of the anisotropic linear anomalous Hall effect (AHE) in the spinel oxide $NiCo_2O_4$ epitaxial film. The unconventional Hall effect indicates a quadrupole dependence on the in-plane current direction being added to the uniform AHE. Moreover, its sign can be manipulated just by magnetic-field cooling. The anisotropic effect is attributed to an electron nematic state originating from a deformed electronic state owing to an extended magnetic toroidal quadrupole and ferrimagnetic order.

Diverse physical properties in condensed matter systems are represented by tensor objects connecting two or more measurable quantities[1]. According to Neumann's principle, the physical property tensor must be invariant under the system's symmetry operation[2,3]. In magnets, the magnetic point group, particularly, the tensor transformation under the time reversal operations ($\mathcal{T}$), should be considered. As for the conductivity of magnetically ordered materials, an electric field **E** induced by an applied electric current with density **J** is written as $E_j = \rho_{ij}(\mathbf{H}, \boldsymbol{\Sigma})J_i$. The electric resistivity tensor $\rho_{ij}$ explicitly expresses external magnetic field **H** and spin configuration $\boldsymbol{\Sigma}$[4–6]. Concerning the $\mathcal{T}$ operation on the resistivity tensor, Onsager's theorem[7] gives a reciprocal relation $\rho_{ij}(\mathbf{H}, \boldsymbol{\Sigma}) = \mathcal{T}\rho_{ji}(\mathbf{H}, \boldsymbol{\Sigma}) = \rho_{ji}(-\mathbf{H}, -\boldsymbol{\Sigma})$. Hall resistivity, an antisymmetric component of transverse resistivity (TR) with respect to **H** and **M**, can lead to[8]

$$\rho_{xy}(H_z, M_z) = -\rho_{yx}(H_z, M_z),\qquad(1)$$

indicating that $\mathbf{J} \parallel x(y)$ induces $\mathbf{E} \parallel y(-x)$ in the presence of $H_z$ and $M_z$. Hence, the Hall voltage is isotropic and independent of the direction of **J**; *i.e.* rotational symmetry is preserved within the normal plane to **H** and **M**, as shown in Fig. 1a[9].

In contrast, since a symmetric TR $\rho_{ij}$, an even-function component of resistivity to **H** and **M**, does not always satisfy Eq. (1), it occasionally exhibits anisotropic behaviour depending on the direction of $\mathbf{J}$[5]. An example is the anisotropic electronic state such as an electronic nematic state[10,11]. The symmetric TR will appear when **J** is applied along the off-principal crystallographic axes. Thus, the response is useful for accurately detecting electronic state deformations[12]. Another intriguing property is the spin Hall effect in an anisotropic spin splitting. For instance, the extended magnetic toroidal quadrupole (MTQ) order with $\mathcal{C}_4\mathcal{T}$ symmetry, *i.e.* a combination operation of 90° rotation ($\mathcal{C}_4$) and $\mathcal{T}$, causes a quadrupole (*d*-wave) shape spin splitting[13]. The spin-by-spin anisotropic electronic state will produce an anisotropic spin resistivity tensor within the *xy*-plane, $\rho_{ij}^z(\boldsymbol{\Sigma}) = \rho_{ji}^z(\boldsymbol{\Sigma})$, with applied $\mathcal{T}$-even spin current $\mathbf{J}_s^z$, as shown in Fig. 1c[14–16]. However, MTQ order forbids the charge current of AHE by its symmetry.

From Onsager's theorem, the Hall effect which is an antisymmetric TR for $H_z$ and $M_z$ is isotropic with respect to the current direction, while a symmetric TR can be anisotropic[5]. Indeed, a planar Hall effect where TRs are anisotropic depending on the angle between **J** and **M** shows a symmetric response to **M**. However, here, we show a manifestation of an unconventional anisotropic Hall effect even for charge current in a conical ferrimagnet composed of MTQ, as shown in Fig. 1c. At first glance, the behaviour violates Onsager's theorem but can be elucidated without contradiction by assuming symmetric MTQ upon $M_z$ reversal.

[1]Department of Applied Physics, University of Tsukuba, Tsukuba, Ibaraki 305-8573, Japan. [2]Research Center for Magnetic and Spintronic Materials (CMSM), National Institute for Materials Science (NIMS), Tsukuba, Ibaraki 305-0047, Japan. [3]Center for Science and Innovation in Spintronics (CSIS), Tohoku University, Sendai 980-8577, Japan. [4]Research and Services Division of Materials Data and Integrated System (MaDIS), National Institute for Materials Science (NIMS), Tsukuba, Ibaraki 305-0047, Japan. [5]Tsukuba Research Center for Energy Materials Science (TREMS), University of Tsukuba, Tsukuba, Ibaraki 305-8573, Japan. ✉e-mail: hiroki.koizumi.d7@tohoku.ac.jp; YAMASAKI.Yuichi@nims.go.jp; yanagihara.hideto.fm@u.tsukuba.ac.jp

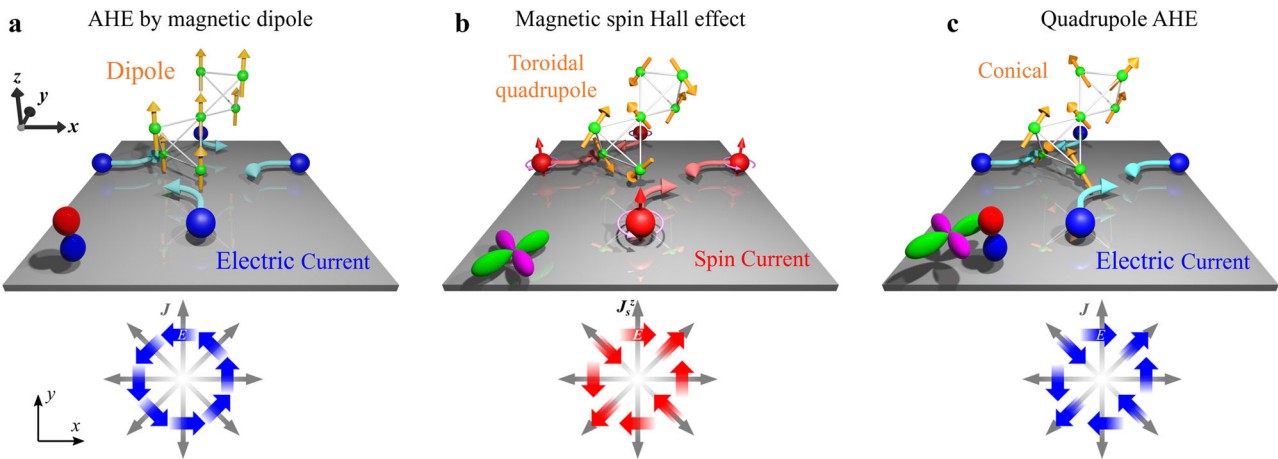

**Fig. 1 | Correspondence among magnetotransport phenomena, centrosymmetric extended magnetic multipoles on pyrochlore lattice, and orbital shapes for magnetic (red and blue) and toroidal (green and magenta) charges.** Isotropic anomalous Hall effect occurs on (**a**) magnetic dipole order (ferromagnet). **b** Extended magnetic toroidal quadrupole order allows anisotropic spin Hall effect but no electric charge Hall effect. **c** Conical magnetic structure comprising the magnetic toroidal quadrupole and magnetic dipole shows anisotropic electric charge Hall effect.

The anisotropic Hall effects were demonstrated on epitaxial $NiCo_2O_4(001)$ films on $MgAl_2O_4(001)$ substrates. $NiCo_2O_4$ is a conductive inverse spinel oxide ($Fd\bar{3}m$) exhibiting ferrimagnetism with a Néel temperature as high as $T_N \approx 400$ K[17]. Stoichiometric thin films exhibit perpendicular magnetic anisotropy at room temperature[18–20]. Moreover, sharp out-of-plane magnetisation switchings are observed in the magnetic field dependence of AHE over the entire temperature range[18,21]. In contrast, the anti-site $Ni^{3+}$ distribution, which can be manipulated by the $O_2$ flow rate during thin film deposition[22], changes the magnetic anisotropy from perpendicular to an easy-cone magnetic anisotropy at low-temperatures[20]. In the present study, we found that such anti-site $NiCo_2O_4$ thin film realises a conical ferrimagnetic structure composed of the out-of-plane ferromagnetic and in-plane MTQ components.

## Results

### Dependence of anomalous Hall effect on the direction of current application

We first show experimental evidence for the anisotropic Hall effect measured at 5 K. Prior to the low-temperature measurements, the sample was cooled under a low positive or negative magnetic field ($\mu_0 H_{FC} = \pm 0.1$ T), apparently perpendicular to the film plane from room temperature. Figure 2a–h show the applied magnetic field **H** ‖ [001] dependence of Hall effects with changing **J** direction. The Hall resistivities $\rho_{ij}^O$ ($i,j = x$ : [100], $y$ : [010], $x'$ : [110], $y'$ : [1$\bar{1}$0]) were extracted by antisymmetrisation analysis (raw data and symmetric TR $\rho_{ij}^E$ are shown in Supplementary Figs. 1 and 2, respectively). The Hall effects measured after a field cooling (FC) procedure with the positive and negative $H_{FC}$ are indicated as $\rho_{ij}^{O+}$ and $\rho_{ij}^{O-}$, shown in Fig. 2a–h, respectively. In the high-$H$ region ($|\mu_0 H| > 1$ T), all Hall resistivity curves behave similarly with a coincident saturation of 5.25 μΩcm. However, they show peculiar behaviour different from the conventional AHE in the low-$H$ region. These results are reminiscent of the topological Hall effect[23,24]; however, they exhibit anisotropic behaviours that are dependent on the direction **J**; $\rho_{x'y'}^{O\pm}$ and $-\rho_{y'x'}^{O\pm}$ are similar to the conventional AHE proportional to $M_z$ [see Supplementary Fig. 8a]. Whereas there are distinct additional contributions in $\rho_{xy}^{O\pm}$ and $-\rho_{yx}^{O\pm}$.

The Hall effects can be decomposed into isotropic and anisotropic components in the applied **J** direction. Figure 2i, j show the anisotropic $\rho_{xy}^{OA+}$ and isotropic $\rho_{xy}^{OI+}$ components derived, respectively, which are half of the difference and average between $\rho_{xy}^{O+}$ and $-\rho_{yx}^{O+}$. The observed largest value of $\rho_{xy}^{OA+}$ is 6.9 μΩcm, which is

~1.3 times that of the saturation value of conventional AHE, as shown in Fig. 2i, j, respectively. In contrast, the isotropic $\rho_{xy}^{OI+}$ seems to coincide with $\rho_{x'y'}^{O\pm}$. Hence, the anisotropic Hall effect has $\cos 2\varphi$ response for the current angle $\varphi$ by the [100] axis. When $H_{FC}$ is inverted, the anisotropic $\rho_{xy}^{OA-}$ is completely sign-reversed within the experimental error, as shown in Fig. 2k. In the case of zero-field cooling (ZFC), the anisotropic Hall effect cancels, and the curve is identical to $\rho_{xy}^{OI\pm}$, indicating that the cooling process does not affect the $M_z$-$H$ curve, as shown in Fig. 2j. To generalise the above results, the series of Hall resistivity can be expressed as

$$\rho^{O\pm}(\varphi) = \rho^{OI} \pm \rho^{OA} \cos 2\varphi \qquad (2)$$

with $\rho^{OI}$ and $\rho^{OA}$ being the isotropic and anisotropic components of Hall effect, respectively. The plus/minus sign indicates the sign of cooling magnetic field [see also Supplementary Fig. 3]. At first glance, such an anisotropic result appears to involve a planar Hall effect caused by assuming the presence of in-plane magnetisation. However, it cannot explain the antisymmetric behaviour with respect to $H_z$ and $H_{FC}$. The possibility of AHE from an extrinsic origin, such as the orientated magnetic domains[25] or phase separation[26], is also excluded in the present sample for the same reason. Hereafter, the anisotropic Hall effect will be referred to as a quadrupole AHE (QuadAHE) to distinguish it from the conventional AHE and the planar Hall effect.

### Temperature dependence of the anisotropic anomalous Hall effect

Since the QuadAHE responds at low-$H$ and is reversed by $H_{FC}$, it is assumed to originate from a nontrivial magnetic structure. We discuss possible magnetic structure coupling with the anisotropic electronic state in the tetragonally distorted spinel structure $NiCo_2O_4$ based on temperature dependence [Fig. 3a,b] and symmetry of resistivities. The spinel structure consists of a diamond lattice at the A-site and a pyrochlore lattice at the B-site (see Supplementary Fig. 7e). Because of the antiferromagnetic interaction between A- and B-sites, $J_{AB}$, the Néel-type collinear ferrimagnetic order is realised with the perpendicular magnetic anisotropy at room temperature[18,19,21]. With decreasing temperature, there is a transition from the Néel-type to a non-collinear magnetic structure with a canted spin at the A-site owing to the change in spin anisotropy from the perpendicular to the easy-cone magnetic anisotropy[20]. Such transformation is identified in the conventional AHE shapes owing to the suppressed **M** around zero magnetic fields, as shown in Fig. 3a. At high temperatures, **M** saturates at low fields, and

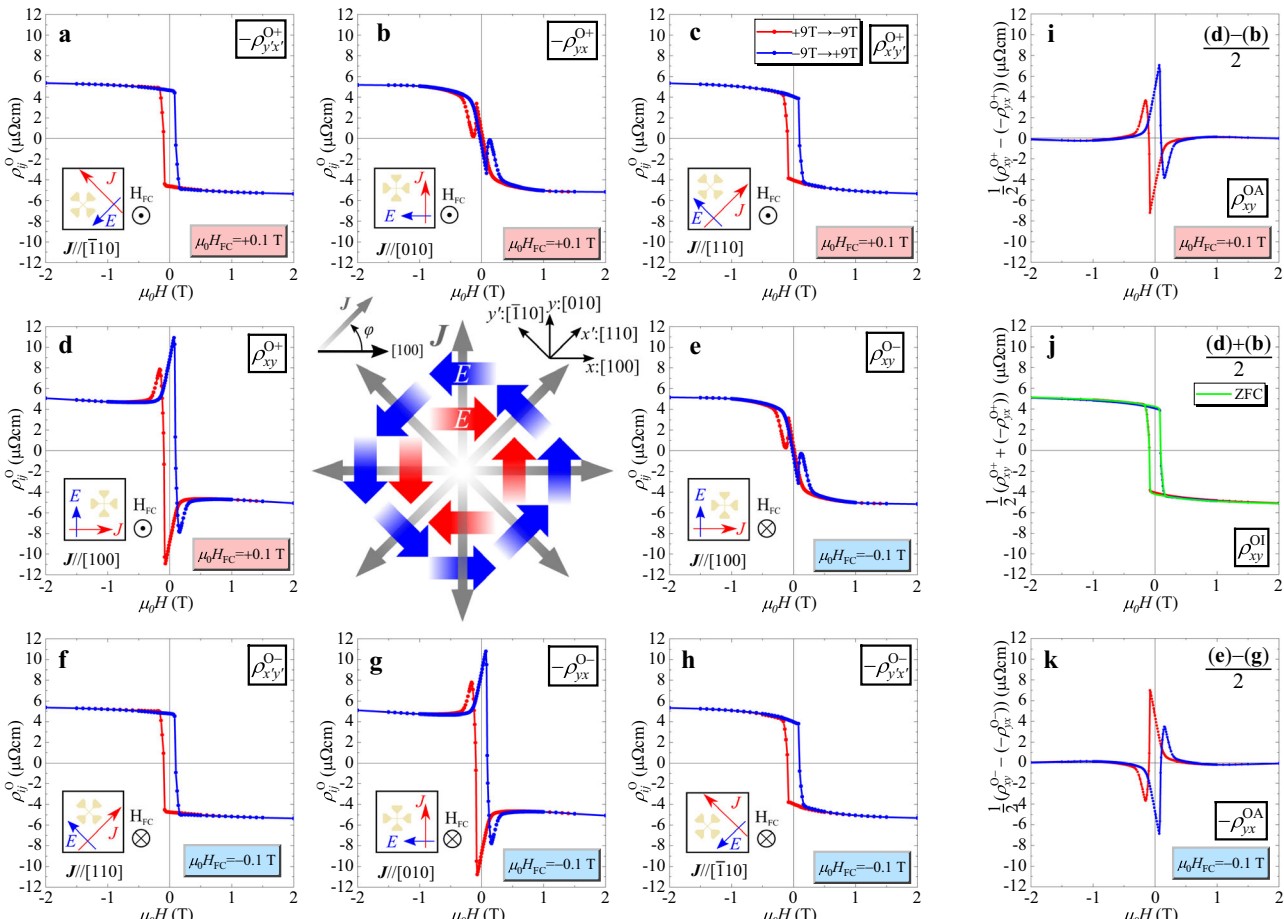

**Fig. 2 | Magnetic field dependence of Hall effect $\rho_{ij}^{O}$, antisymmetric component of transverse resistivity, with changing applied current J direction at 5 K.**
**a–d**, **h**, **g** and **f**, **e** indicate Hall resistivities for $\mathbf{J} \| y' : [\bar{1}10]$, $y : [010]$ $x' : [110]$, and $x : [100]$, respectively, measured after positive (negative) field cooling ($H_{FC}$). The central figure shows the directions of the applied current and induced electric field; blue and red arrows represent the conventional and unconventional TRs observed in the low-field region, respectively. The measurements setup, e.g. Hall bar shape and current direction, is illustrated in the inset schematic of each figure. **i** Extracted anisotropic and (**j**) isotropic components of $\rho_{ij}^{O}$ for $\mu_0 H_{FC} = +0.1$ T. **k** Anisotropic component for $\mu_0 H_{FC} = -0.1$ T.

the remanence is almost identical to the saturation but decreases at lower temperatures. Fig. 3c shows the temperature dependence of the squareness ratio defined as the ratio of AHE remanence to saturation. It is 1 in the higher temperature collinear phase and declines at temperatures below the magnetic transition temperature $T_S \approx 130$ K. Concurrent with the magnetic transformation, the symmetric TR appears below $T_S$, as shown in Fig. 3d [see also Supplementary Fig. 4]. The symmetric and anisotropic TR, $\rho_{xy}^{E}(H) = \rho_{yx}^{E}(H)$, as seen in Supplementary Fig. 2, suggests that the electronic state is anisotropic within the (001) plane. However, it is unaffected by the reversal of $H_{FC}$ and ascribed to different origins from the QuadAHE. This is presumed to be an electronic state change originating from a coplanar magnetic structure, the so-called Yafet-Kittel-type two-dimensional magnetic structure with canted magnetic moment on the A-site[27]. The symmetric and anisotropic TR $\rho_{xy}^{E}$ can be regarded as the planar Hall effect originating from the in-plane antiferromagnetic component of the coplanar Yafet-Kittel-type magnetic structure. In contrast, the QuadAHE emerges below $T_Q \approx 80$ K, as shown in Fig. 3b,d, showing variations of $\rho_{xy}^{OA}$ and temperature dependence of remanence. There is no apparent anomaly in the squareness ratio and symmetric TR at $T_Q$, hence the QuadAHE is attributed to a further magnetic transformation from the Yafet-Kittel-type magnetic structures on the B-site at $T_Q$. Such a magnetic structure change from collinear to multistep non-collinear has not been observed in stoichiometric samples[18,21,28]. The results suggest that the easy-cone magnetic anisotropy due to the anti-site

Ni$^{3+}$ distribution in the present sample is essential for structural changes.

## Discussion
### Magnetic structure based on the symmetry of anomalous Hall effect
At high temperatures and high-$H$ collinear ferrimagnetic states, the B-site magnetic structure belongs to the magnetic point group of $4/mm'm'$, indicating an isotropic electronic state within the film plane (Supplementary Table I), consistent with the conventional AHE. In contrast, the QuadAHE, in Eq. (2), in the low-$H$ region suggests a magnetic structure with $C_4\mathcal{T}$ symmetry as in the in-plane antiferromagnetic component. Such a structure is realised in a pyrochlore lattice composed of the spinel B-site. There are 12 orthonormal irreducible magnetic structures in the pyrochlore lattice characterised by extended multiples of spatial inversion symmetry, namely three magnetic dipole, four magnetic octupole and five MTQ moments (see Refs. [29,30]). The $T_{1g}$ magnetic octupole with $4/mm'm'$ magnetic point group would induce AHE, whereas the $A_{2g}$ octupole, known as the all-in-all-out structure, will not induce AHE due to its $m\bar{3}m'$ symmetry. However, bi-axial epitaxial strain in thin films can make the magnetic octupole component nonzero in the all-in-all-out structure, thus inducing AHE[31]. In the present epitaxial in-plane tensile strain, there are four possible in-plane antiferromagnetic bases, MTQ $T_v$ $(4/mmm)$, MTQ $\tilde{T}_u$ $(4'/mmm')$, MTQ $T_{xy}(4'/mm'm)$, and magnetic octupole

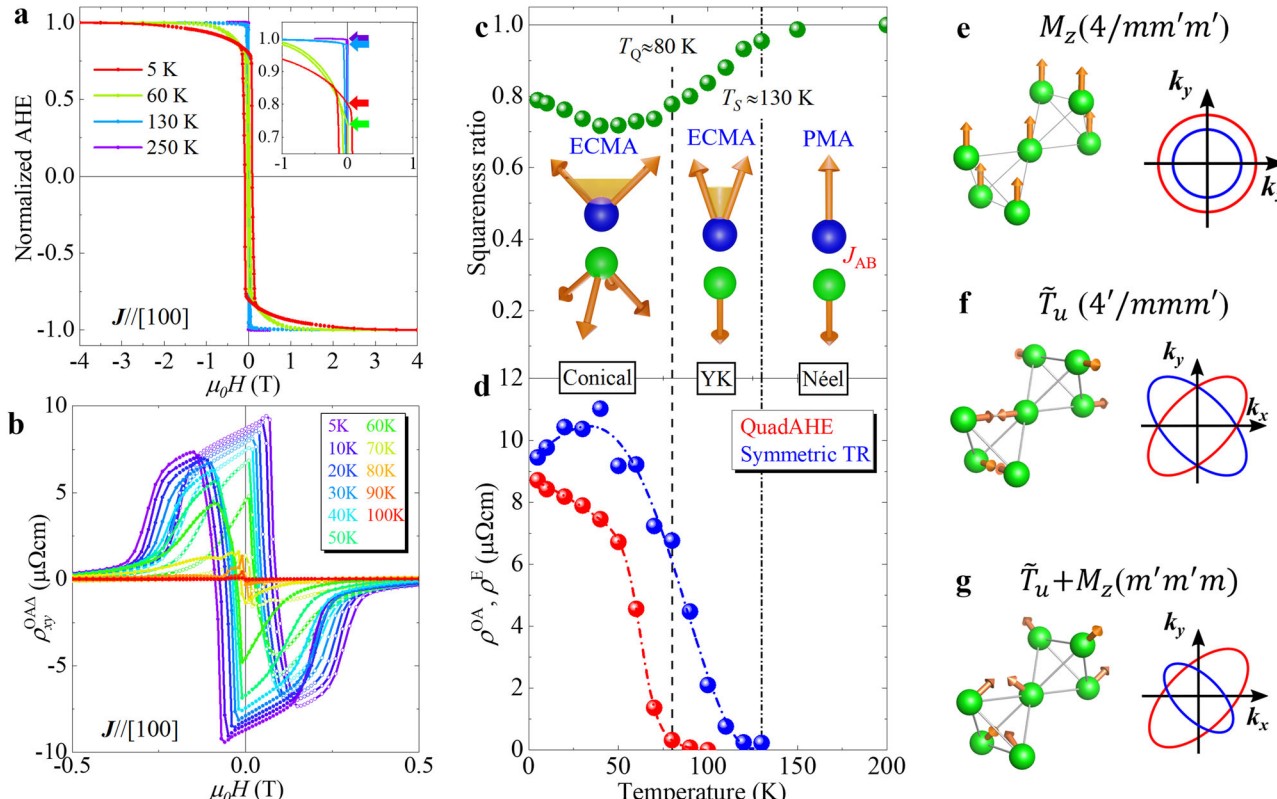

**Fig. 3 | Temperature dependence of anomalous Hall effect (AHE) and electronic states modulated by magnetic order.** Magnetic field dependence of (**a**) normalised AHE after ZFC and (**b**) extracted anisotropic components for $\mathbf{J} \parallel [100]$ measured at selected temperatures. The inset of (**a**) indicates the ratio of saturation and remanence of AHE, defined as a squareness ratio. Temperature dependence of **c** squareness ratio and **d** symmetric TR and anisotropic Hall effect, $\rho^E$ and $\rho^{OA}$,

respectively. The insets of (**c**) indicate the assumed magnetic structure for the A- and B-sites, drawn by blue and green spheres, respectively. Magnetic structure on pyrochlore lattice and band dispersion for (**e**) magnetic dipole $M_z$ [$4/mm'm'$], (**f**) magnetic toroidal quadrupole $\tilde{T}_u$ [$4'/mmm'$], and (**g**) conical magnet $M_z + \tilde{T}_u$ [$m'm'm$].

$M_z^\alpha (4/mm'm')$, as shown in Supplementary Fig. 5a. Here, MTQ $\tilde{T}_u$ corresponds to a linear combination of MTQ $T_u$ and magnetic octupole $M_{xyz}$, as shown in Fig. 3f. Among these four components, $4'/mm'm$ or $4'/mmm'$ possibly elucidate QuadAHE because of $\mathcal{C}_4\mathcal{T}$ symmetry. Considering the existence of the (110) mirror symmetry with $\mathcal{T}$ in QuadAHE, MTQ $\tilde{T}_u$ ($4'/mmm'$) reasonably accounts for the experimental results.

$\mathcal{T}$-odd magnetic orders with spatial inversion cause symmetric band dispersions with spin splitting[13]. The band dispersion of the ferromagnet, i.e. the magnetic dipole order, shows a uniform spin polarised structure, as shown in Fig. 3e. The $\mathcal{T}$-odd MTQ $\tilde{T}_u$ moment $4'/mmm'$ demonstrates the quadrupole-type ($d$-wave like) spin splitting; the up and down spin bands are elliptical and rotate at 90° from each other, as shown in Fig. 3f. In the pure MTQ antiferromagnetic order, the spin Hall effect is allowed[16], but the electric charge Hall effect is prohibited owing to $\mathcal{C}_4\mathcal{T}$ symmetry. In contrast, in a conical magnetic structure consisting of magnetic dipole and MTQ, $\mathcal{C}_4$ symmetry is broken. The charge and spin Hall effects appear because of the anisotropic Fermi surface, namely the liquid crystal-like electron nematic state, originating from the differences in the spin density, as shown in Fig. 3g.

**Magnetotransport model for QuadAHE.** To approach the magnetotransport phenomena, we consider a minimal model Hamiltonian in the MTQ $\tilde{T}_u$ conical magnetic order written as

$$\mathcal{H} = \sum_{\mathbf{k}\sigma\sigma'} \left[ \frac{\hbar^2}{2m} \left( \mathbf{k}^2\sigma_0 + 2\tilde{t}_u k_x k_y \sigma_z \right) + m_z\sigma_z \right] c_{\mathbf{k}\sigma}^\dagger c_{\mathbf{k}\sigma'} \quad (3)$$

where $c_{\mathbf{k}\sigma}^\dagger$ ($c_{\mathbf{k}\sigma}$) is the creation (annihilation) operator of an electron with wave vector $\mathbf{k}$ and spin $\sigma_i$ ($i = 0, x, y, z$)[13]; $m_z$ and $\tilde{t}_u$ denote the molecular fields from the magnetic dipole $M_z$ and MTQ $\tilde{T}_u$ orders, respectively. Here, we only consider the $k_z = 0$ plane for simplicity. $k_x k_y \sigma_z$ corresponds to the quadrupole-type spin splitting due to the $\mathcal{C}_4\mathcal{T}$ symmetry. Then, Boltzmann's transport equation gives the charge conductivity as

$$\begin{pmatrix} \sigma_\parallel \\ \sigma_\perp \end{pmatrix} = \sigma_D \begin{pmatrix} \bar{n}(1 + \tilde{t}_u^2) + 2\Delta n \tilde{t}_u \sin 2\varphi \\ 2\Delta n \tilde{t}_u \cos 2\varphi \end{pmatrix}, \quad (4)$$

with the average ($\bar{n}$) and difference ($\Delta n$) of electron number for up and down spins. Assuming $|\sigma_\perp| \ll |\sigma_\parallel|$, the anisotropic resistivities are obtained as

$$\rho = \begin{pmatrix} \rho_\parallel \\ \rho_\perp \end{pmatrix} = -\frac{2\eta}{\sigma_D^2} \begin{pmatrix} \sin 2\varphi \\ \cos 2\varphi \end{pmatrix}, \quad (5)$$

where $\eta = \tilde{t}_u \Delta n$ and $\sigma_D$ is isotropic longitudinal conductivity. The $\cos 2\varphi$ response reproduces the angle dependence of the observed QuadAHE. Since $\Delta n$ and $\tilde{t}_u$ are $\mathcal{T}$-odd, $\rho_\perp$ is proportional to the $\mathcal{T}$-even quantity $\eta$. If $\tilde{t}_u$ is not reversed with $M_z$ reversal, namely preserving the in-plane antiferromagnetic structure while reversing the perpendicular ferromagnetic component $M_z$, QuadAHE exhibits an antisymmetric response with respect to $M_z$. Figure 4a shows schematics of the electronic bands and magnetic structure modified by the magnetic field change. At high magnetic fields, the band dispersion is isotropic owing to the collinear ferrimagnetic structure; in contrast, under weak magnetic fields, the MTQ $\tilde{T}_u$ conical magnetic structure contributes to

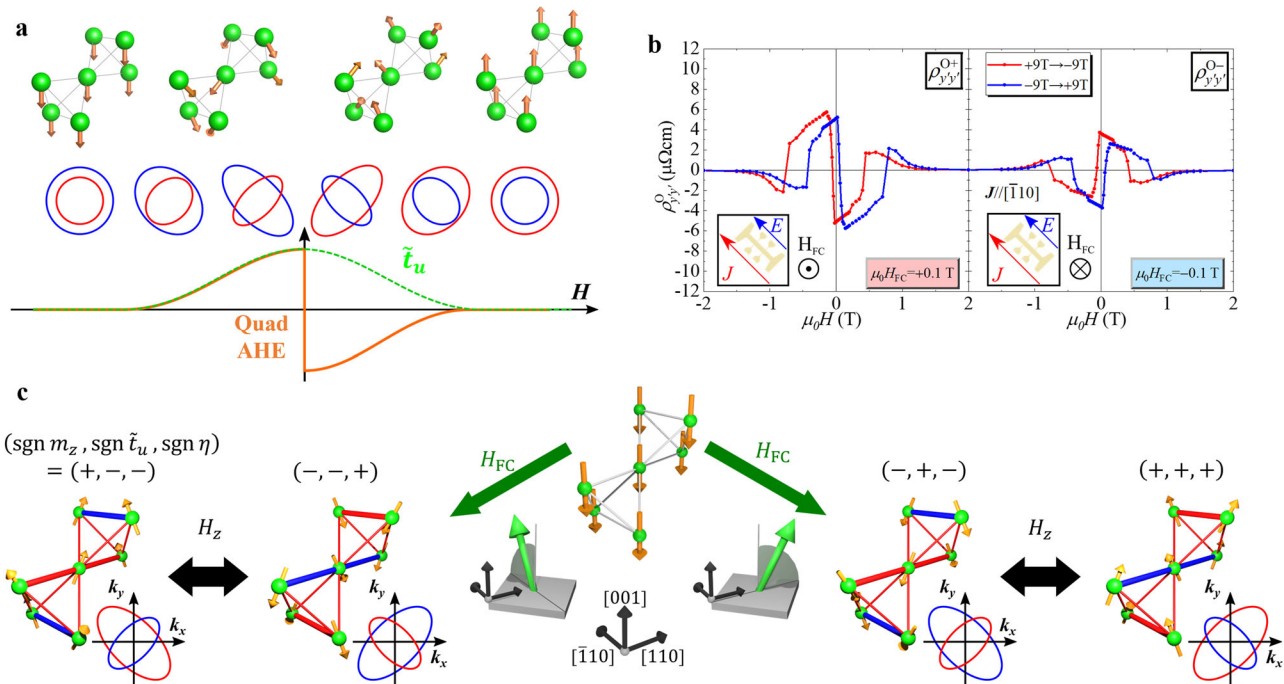

**Fig. 4 | Schematics of extented magnetic toroidal quadrupole conical order dependent on magnetic field. a** Assumed magnetic field dependence of magnetic structure, electronic band structure, and corresponding anisotropic Hall effect. **b** Magnetic field dependence of antisymmetric longitudinal magnetoresistance with **J** ∥ [$\bar{1}$10] and **H** ∥ [001]. **c** Relations of the four degenerated conical magnetic structures and magnetic fields. The signs of $m_z$ and $\tilde{t}_u$ are determined by the $z$ component and the direction of the in-plane component of the magnetic field $H_{FC}$ applied upon the magnetic transition from the B-site collinear to the MTQ conical magnetic structure. After realising the single domain MTQ conical magnetic structure, the isothermal magnetic field reversal inverts $M_z$ and not MTQ $\bar{T}_u$.

the anisotropic band structure. Assuming symmetric MTQ $\tilde{T}_u$ with respect to $M_z$, only the band sizes of the up and down spins change, contributing to QuadAHE reversal. This model also suggests that the transverse magnetoresistance (MR) effect, i.e. change of longitudinal resistance by perpendicular **H**, exhibits an antisymmetric MR effect to $M_z$[32-34] and an anisotropic response to applied electric current. Figure 4b indicates the magnetic field dependence of antisymmetrised MR along the $y' : [\bar{1}10]$ direction ($\varphi = 135°$). The antisymmetric resistivity, $\rho_{y'y'}^{O+}(-H_z, -M_z) \approx -\rho_{y'y'}^{O+}(H_z, M_z)$, can be recognised in the low-$H$ region, and its sign is reversed by the reversal of $H_{FC}$, i.e. $\rho_{y'y'}^{O-}(H_z, M_z) \approx -\rho_{y'y'}^{O+}(H_z, M_z)$ and QuadAHE.

**Magnetic domain control of extended MTQ moment by magnetic-field cooling.** The QuadAHE is fully sign-reversed by the inversion of $H_{FC}$ [Fig. 2i,k], indicating the absence of complexities, such as multiple MTQ domains. During the isothermal reversal of $M_z$, the MTQ $\tilde{T}_u$ single-domain feature, realised using the magnetic-field cooling procedure, was nearly maintained. It suggests that MTQ and magnetic dipole form independent domain walls without coupling. These behaviours are in contrast with conical magnet multiferroics, wherein electric and magnetic fields are required to realise the single domain and an inversion of spin helicity occurs with the reversal of $M_z$[35,36]. Figure 4c depicts four possible MTQ $\tilde{T}_u$ conical magnetic structures, accessed using $H_{FC}$ and isothermal reversal of $H_z$ with respect to the sign of $\tilde{t}_u$, $m_z$, and $\eta$. The directional alignment of electron nematic state can be manipulated by $M_z$ reversal due to the symmetric property of MTQ.

MTQ selection by magnetic field cooling can be interpreted by the Dzyaloshinskii-Moriya (DM) interactions[37,38]. In the pyrochlore structure, a nonzero DM vector exists on bonds between the B-site ions[39,40]. For every bond, the magnitude and sign of DM interaction energy in the MTQ conical magnetic order are indicated by its radii and colours, respectively. The four MTQ conical states energetically degenerate if the conical axis is parallel to the $z$-axis. However, for non-equivalent magnitudes of DM vectors owing to the epitaxial strain, the

degeneracy is lifted when the conical axis is tilted toward the ⟨110⟩ axes (details are provided in Supplementary Note 8). The most stable MTQ conical state is uniquely determined by the sign of $M_z$ and the tilting direction ([110] or [1$\bar{1}$0]). Namely, the single MTQ domain is realised by slightly tilted $H_{FC}$ applied on the magnetic structure transition during cooling, and its sign is determined by the tilted direction and sign of $H_z$. In the actual experiment, we apply $H_{FC}$ in the $z$ direction; however, the in-plane magnetic field component may be attributed to the mis-alignment of partially unadjusted equipment. This conjecture is supported by the fact that the selected MTQ sign reverses when rotating the sample by 90° in the experimental arrangement corresponding to the rotation of the in-plane $H_{FC}$ component between the [110] and [1$\bar{1}$0] axis [Supplementary Fig. 6].

The easy-cone magnetic anisotropy and DM interaction manifest the magnetic structural change from ferromagnetic to MTQ conical magnetic structure on pyrochlore lattice in the anti-site NiCo$_2$O$_4$ thin film. Since the energy scales of magnetic anisotropy and DM interactions are weaker than the antiferromagnetic superexchange interactions[41], QuadAHE is suppressed at lower magnetic fields compared to the AHE in frustrated magnetic pyrochlore materials[8,42,43]. However, the experimental results confirm the existence of the same QuadAHE curve even after returning from a higher magnetic field of 9 T to a lower magnetic field (<0.3 T). Upon applying a magnetic field, the MTQ conical magnetic structure gradually changes with decreasing cone angle; however, it could not become completely collinear due to the influence of the remaining easy-cone magnetic anisotropy, preserving stable MTQ sign information.

**Consideration by Onsager's reciprocal theorem**
Finally, we reconsider the Onsager reciprocal theorem in the QuadAHE. Within the same sign of $H_{FC}$, the observed Hall resistivity $\rho_{ij}$ satisfies the Onsager reciprocal relation of Eq. (1) when $(i,j) = (x', y')$ but not when $(i,j) = (x,y)$. However, considering the $\mathcal{T}$-odd MTQ $\tilde{T}_u$ whose sign is determined by that of $H_{FC}$, the Onsager reciprocal theorem can

be satisfied by extending $\rho_{ij}(\mathbf{H}, m_z, \tilde{t}_u) = \rho_{ji}(-\mathbf{H}, -m_z, -\tilde{t}_u)$. In other words, the Onsager reciprocal theorem is certainly satisfied for the $\mathcal{T}$ operation which reverses all the spin directions. When the Hall resistivity is $\mathcal{T}$-odd, it should be isotropic such as that in the conventional AHE. From contrapositive reasoning, the manifestation of anisotropic QuadAHE requires a term proportional to the $\mathcal{T}$-even quantity. This would correspond to the interference term $\eta$, as in Eq. (5). Therefore, from the point of Onsager reciprocal theorem, the existence of symmetric MTQ to $M_z$ reversal is mandatory for antisymmetric QuadAHE. For the longitudinal MR, the Onsager reciprocal theorem indicates that $\rho_{ij}(\mathbf{H}, \boldsymbol{\Sigma}) = \rho_{ij}(-\mathbf{H}, -\boldsymbol{\Sigma})$, suggesting that the antisymmetric MR is forbidden. However, the existence of the symmetric MTQ makes antisymmetric MR admissible.

In summary, we unveiled the unconventional AHE and MR, that is antisymmetric with respect to $M_z$ and anisotropic Hall effect depending on the applied current direction. The results are explained by the electron nematic state induced by the quadrupolar spin-split band structure and energy shift due to MTQ and magnetic dipole order, respectively. The MTQ conical can be manipulated by isothermal field inversion and magnetic-field cooling. Though the anisotropic Hall effect and antisymmetric magnetoresistivity seem to violate the Onsager reciprocal theorem, it can be understood considering a resistivity proportional to $\mathcal{T}$-even interference terms between the MTQ and magnetic dipole, i.e. $\eta = \tilde{t}_u \Delta n$, which is symmetric with $\mathcal{T}$ operation but antisymmetric with respect to $M_z$. The degeneracy of the conical MTQ structure can be lifted by the tilted magnetic field applied on cooling due to the difference in the DM interaction by the thin film epitaxial compressive strain.

The results are expected to pave the way for new emergent properties with potential applications to spintronic devices such as a multivalued memory device using the electron nematic state and can open a new research field of the Hall effect originating from extended magnetic multipoles. Although the present study infers the realisation of electron nematic state[44–46] from QuadAHE, for example, an angle-resolved photo-emission spectroscopy measurement will provide more direct evidence of distorted Fermi surface and clarify a different origin than the nematic state observed in $Sr_3Ru_2O_7$ and other systems[47,48]. The MTQ conical magnetic structure of $NiCo_2O_4$ is not directly determined in the current study. Since it is a thin-film sample, it is difficult to analyse the magnetic structure by neutron scattering which requires the volume of the sample. In addition, since the magnetic anisotropy depends on the anti-site $Ni^{3+}$ distribution, the specific growth conditions under which the MTQ conical magnetic structure realises have not yet been found. Additional research is required to gather more evidence that will further support the present model.

## Methods

### Sample preparation and measurements
Epitaxial $NiCo_2O_4$ films with 50 nm thickness were grown on $MgAl_2O_4$ substrate by the reactive RF magnetron sputtering technique (ES-250MB: Eiko Engineering Co. Ltd.). We used a 2-inch alloy target with a nominal composition of Ni:Co = 1:2. The growth conditions of $NiCo_2O_4$ films were Ar and $O_2$ flow rates of 10 and 5.0 sccm, respectively, a process temperature of 300 °C, and a working pressure of 1.5 Pa. Finally, we cooled the $NiCo_2O_4$ film to room temperature under an oxygen pressure of 0.8 Pa.

For investigating electric properties, the film was patterned into Hall bars by photolithography and Ar ion milling. Next, Cr and Au electrode layers were sputtered for longitudinal resistivity (LR) and TR measurements. The shapes of Hall bars are shown in each figure (Fig. 2 and Supplementary Fig. 3) and have sizes of 20 μm × 300 μm and 200 μm × 1400 μm. The different devices/electrodes are isolated by the Ar ion milling technique. Since the substrate $MgAl_2O_4$ is an insulator, each hole bar is electrically separated by removing the $NiCo_2O_4$ film other than the hole bar by the milling. The electric properties were measured with a physical property measurement system using a DC

current source (Keithley 6221) and nanovoltmeter (Keithley 2182). For measuring the anisotropy of TR, electric currents are applied along $\mathbf{J} \parallel [100]$, $\parallel [110]$, $\parallel [010]$, and $\parallel [\bar{1}10]$.

## Data availability
The data that support the findings of this study are available from the corresponding author upon request.

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

## Acknowledgements

The authors thank T. Arima and H. Nakao for the productive discussion. Microfabrication of this work was carried out under the MEXT program of Advanced Research Infrastructure for Materials and Nanotechnology in Japan (JPMXP1222BA0017) and the cooperation of T. Kashiwagi and Nanotechnology Platform at the University of Tsukuba. This work was performed under the approval of the "Photon Factory Program Advisory Committee" (proposals No.2017G602 and No. 2016S2-005). H.K. thanks Y. Hatsugai, S. Kuroda and S. Mitani for their useful comments. This project is partly supported by the Japan Society for the Promotion of Science (JSPS) KAKENHI (23H01842: H.Y., 22H04966: H.Y., 21H01750:H.Y. and 19H04399: Y.Y.) and TIA-Kakehashi (TK22-023: H.Y. and TK23-017: H.Y.). This work is also partially supported by PRESTO (JPMJPR177A: Y.Y.) and CREST(JPMJCR1861: Y.Y.), Japan Science and Technology Agency (JST). H.K. acknowledges partly supports of Grant-in-Aid for JSPS Fellows (20J10749:H.K. and 22J00871:H.K.).

## Author contributions

H.K. fabricated thin films, performed experiments, and collected data. H.K., Y.Y. H.Y. discussed the results, wrote the paper, and prepared figures.

## Competing interests

The authors declare no competing interests.
