## [Peer Review File · Nature Communications]

REVIEWER COMMENTS

Reviewer #1 (Remarks to the Author):

In this manuscript, Prof. Yanagihara and colleagues reported an anisotropic unconventional Hall effect induced by electron liquid crystal state, arising from the toroidal quadrupole conical magnetic order of NiCo₂O₄ film. NiCo₂O₄ possesses extraordinary electrical, magnetic and electrochemical properties, has attracted extensive interest and shown excellent application prospects in various fields. The results of this manuscript are interesting. I expect that the authors could answer following questions:

(1) In the manuscript, the authors compared NiCo₂O₄ with pyrochlore magnets and attributed the QuadAHE to peculiar magnetic structures at pyrochlore sites. According to previous reports, the two-in-two-out or one-in-three-out spin structures in A₂B₂O₇ remain stable with magnetic field up to 7 Tesla. The related enhancement or degeneration of AHE also hold in the whole measured field region [1-2]. In contrast, the QuadAHE only appears at low field. What is the mechanism behind the different Hall behaviors between NiCo₂O₄ and pyrochlore magnets?

[1] Y. Taguchi, et al., Science 291, 2573 (2001).

[2] Y. Machida, et al., Phys. Rev. Lett. 98, 057203 (2007).

(2) In the manuscript, the QuadAHE is triggered by field-cooling process under a low magnetic field of ± 0.1 Tesla. The transverse resistivity was measured under the field up to 4 Tesla (see Fig. 3a and Supplementary Fig. 2). All spins at the pyrochlore sites might be aligned along the field direction at such a high field. What is the effect of the applied 0.1 Tesla during the field-cooling process under this situation? Did authors consider the effect of magnetic domains, which may play a dominant role in magnetization and AHE at low magnetic fields. Is it possible that the orientated magnetic domains[3] or phase separation[4] are enhanced during the process of field-cooling at ± 0.1 Tesla, then cause the anisotropic transverse resistivity at low temperature? The Hall results in the literature show the similar anomaly at low field. [4]

[3] Joonghoe Dho, Jungbae Kim, Thin Solid Films 756, 139361 (2022).

[4] M. Xue et al., ACS Appl. Electron. Mater. 2, 3964–3970 (2020).

(3) According to $\rho_{xy} = R_0 H + S_A \rho_{xx}^n M$, magnetization generally have influence on the transverse resistivity. Consequently, the anisotropic M vs. H curves with both field-cooling and zero-field-cooling processes should be presented.

(4) The behavior of field dependence of even-parity transverse resistivity(see Fig. 2 of Supplemental Material) is similar to that of magnetoresistance reported in previous literature[5]. Could the even-parity resistivity be caused by the magnetoresistance signal of Hall electrodes? Did authors consider the effect of Hall electrodes misalignment and the inhomogeneity of films?

[5] X. Xu et al., J. Appl. Phys. 132, 020901 (2022).

(5) In the manuscript, the authors claimed a magnetically induced electron liquid crystal state and cited the Ref. 17. Whether the electronic liquid crystal phase in NiCo₂O₄ and the new phase in Sr₃Ru₂O₇ (Ref. 17) have the same formation mechanism? Is the electron liquid state in NiCo₂O₄ the same as nematic phase observed in other strongly correlated electron systems, for instance high-temperature superconductors.

(6) Authors showed the temperature dependence of odd-parity Hall effect (ρ_{OA}) in Fig. 3d. What is the angular-dependence of ρ_{OA} on the angle between current and [100] axis? Could authors plot a similar figure as the Fig. 3 in Ref. 18 [6]?

[6] J. Wu et al., Nature 547, 432 (2017).

Reviewer #2 (Remarks to the Author):

In this paper, Koizumi et al. present a comprehensive study of anomalous Hall effect in epitaxial NiCo₂O₄ films deposited on (001) MgAl₂O₄ substrates. The authors observe a topological-Hall-like feature in AHE switching hysteresis in field-cooled samples, which is absent in zero field cooling. This unconventional Hall signal shows a quadrupole dependence on the in-plane current direction and flips sign upon reversing the direction of the cooling field. Based on symmetry arguments, the authors attribute the anisotropic Hall effect to the emergence of an electron liquid crystal (ELC) state originating from an extended toroidal quadrupole (MTQ) conical magnetic order.

Overall, the observed anisotropic AHE in NCO is very interesting, but it is not an intrinsic property of NCO rather due to antisite disorder of Ni³⁺, making it less controllable. The proposed scenario of emergent ELC and MTQ order does not have solid experimental support and is far-fetched. The data may be explained by far more trivial transport model. As a result, I cannot recommend the publication of the paper in Nature Communications in the present form. Below are some detailed comments.

1. The paper lacks some basic sample information. Please provide the film thickness of the two samples studied, surface morphology (e.g., AFM image and surface roughness), device details, e.g., whether it is measured via Hall bar or van der Pauw geometry, the type of metal electrodes, device fabrication/lithographic process, etc.
2. Please note that AHE in NCO has been reported in previously studies (e.g., Chen et al., *Adv. Mater.* 31, 1805260 (2019); Kan et al., *PRB* 104, 134407 (2021); Chen et al., *APL* 120, 242401 (2022)). Please cite the relevant work properly and discuss the current work in the context of existing literature.
3. The authors attribute the anisotropic AHE to the transitions from PMA to two types non-collinear spin structures. As discussed in Ref. [26], the appearance of easy cone magnetic anisotropy is not intrinsic to NCO but originates from the antisite distribution of Ni³⁺ occupying the Td site. For example, the authors use the squareness of the AHE hysteresis to gauge the onset of spin canting and ECMA. However, for high quality NCO films on MAO, the square-shape hysteresis is sustained down to the lowest temperature (e.g., 2 K in Chen et al., *Adv. Mater.* 31, 1805260 (2019); Chen et al., *APL* 120, 242401 (2022)) in a wide range of film thickness. The fact that the QuadAHE varies from sample to sample (mentioned in Supplementary) further attests to the extrinsic nature of the effect. Such discussion is absent in the introduction/discussion of the manuscript. Please elaborate on this point in the main text so that the readers are properly informed.
4. In Fig 1a-d, the combination of 3D device schematic and 2D J-E relation is very confusing. Please better separate these two schematics and add corresponding laboratory coordinates.
5. Please define angle phi in Eq. (2) and add it to the schematic in Fig. 2.
6. The main problem of the paper is there is no substantial experimental evidence of the emergent ELC and MTQ order other than the angular dependence of AHE. The angular dependence in Eq. 2 can be simply due to the planar Hall effect (Li et al., *J Vac Sci Technol A* 40, 010807 (2022)), which can

appear in the presence of in-plane magnetization (due to spin-canting, not toroidal magnetic order). It naturally explains the sign change of Hall signal upon reversing the cooling field, as switching the out-of-plane magnetization can reverse the spin canting direction. The planar Hall signal is also suppressed at high perpendicular field as the in-plane magnetization is quenched. This means the relevant physics may be the in-plane magnetocrystalline anisotropy and spin-orbit interaction.

Reviewer #3 (Remarks to the Author):

The authors have reported the anisotropic Hall effect by systematically analyzing transverse resistivities with respect to applied current directions. They have also discussed the origin of anisotropy based on cluster (extended) multipoles which characterize magnetic structures according to the crystallographic symmetry. The experiment and its theoretical analyses appear sound, but it is quite unclear whether their findings are really significant in this and related fields. The main reason of unclearness is quite low readability of the manuscript that contains many inaccurate jargons, and undefined words. Therefore, I do not recommend this work as it is published in Nature Communications. I give several points below that seem to enhance low readability with quite inaccurate use of technical terms, and they make the essential statement of this work obscure. The presentation of the manuscript gives an impression that the work is not well organized and summarized.

(1) In the abstract, the authors use "topological Hall effect" for systems with any of nontrivial magnetic structures. However, this is quite crude and a topological Hall effect has its own meaning and definition. Similarly, "electron liquid crystal state" is not well established, and the present system is just a deformed band structure from conical magnetic ordering, and has nothing to do with "liquid crystal".

(2) The Onsager reciprocal relations are the statement about the off-diagonal components of the resistivity in this context as the authors exhibit in the sentence above Eq.(1), and it does not assume the antisymmetric property with respect to H and M as Eq.(1). Therefore, it is not surprising that the in-plane anisotropy of resistivity tensor appears provided the symmetric components appear in the tensor, which gives rise to anisotropic responses in total.

(3) The authors use the terms, even-parity and odd-parity, for Hall effect, which are quite unclear. Moreover, the word "Hall effect" is usually used to refer to the antisymmetric components in the resistivity tensor, since the symmetric components can be eliminated by appropriate principal-axis transformation. The main message of this manuscript seems to be the existence of both symmetric and antisymmetric components due to the conical spin structure accompanying (cluster) magnetic dipole and magnetic toroidal quadrupoles, however, it may be smeared out by inaccurate use of words.

(4) The condition of disappearance of anomalous Hall effect is not the same as no (zero) Berry curvature, since the former is determined by the sum of the latter over the occupied electronic states. However, there are several inaccurate statements in the manuscript.

(5) The authors use "Mz-even" and "Mz-odd" and so on, in which the use of even/odd is quite unclear.

(6) In my opinion, the authors quite frequently use their own words without definitions, and they are not familiar to the readers in the related fields. I should strongly recommend to improve sentences and English expressions.

Reviewer #1 (Remarks to the Author):

[Comment]

In this manuscript, Prof. Yanagihara and colleagues reported an anisotropic unconventional Hall effect induced by electron liquid crystal state, arising from the toroidal quadrupole conical magnetic order of NiCo₂O₄ film. NiCo₂O₄ possesses extraordinary electrical, magnetic and electrochemical properties, has attracted extensive interest and shown excellent application prospects in various fields. The results of this manuscript are interesting. I expect that the authors could answer following questions:

[Response]

We thank the reviewer for evaluating our manuscript positively and giving constructive suggestions, which are all helpful in improving the present manuscript. Below are the point-by-point replies to the reviewer's comments and concerns.

[Comment 1]

(1) In the manuscript, the authors compared NiCo₂O₄ with pyrochlore magnets and attributed the QuadAHE to peculiar magnetic structures at pyrochlore sites. According to previous reports, the two-in-two-out or one-in-three-out spin structures in A₂B₂O₇ remain stable with magnetic field up to 7 Tesla. The related enhancement or degeneration of AHE also hold in the whole measured field region [1-2]. In contrast, the QuadAHE only appears at low field. What is the mechanism behind the different Hall behaviors between NiCo₂O₄ and pyrochlore magnets?

[1] Y. Taguchi, et al., Science 291, 2573 (2001).

[2] Y. Machida, et al., Phys. Rev. Lett. 98, 057203 (2007).

[Response 1]

We appreciate the reviewer's insightful comments regarding the magnetic field response of conical magnetic structures. In the pyrochlore lattice, non-collinear magnetic structures, such as the 2-in-2-out or the 1-in-3-out configurations, are realised due to antiferromagnetic exchange interactions and geometric frustration. Conversely, spin frustration is suppressed in NiCo₂O₄ due to its antiferromagnetic exchange interactions between the A and B sites. While the bulk crystal exhibits ferromagnetism with parallel spin alignment, NiCo₂O₄ thin-film crystal undergoes epitaxial distortion, leading to changes in magnetic anisotropy and the Dzyaloshinskii-Moriya interaction (DMI). In particular, A₂B₂O₇ and NiCo₂O₄ employ different mechanisms to form non-collinear magnetic structures. The former relies on the exchange interaction with an energy scale of 10 meV, whereas the latter involves the DMI or magnetic anisotropy at the order of 1 meV. This discrepancy in magnetic energy contributes to the distinct response of the conical

magnetic structure to an external magnetic field. Notably, the magnetic field required to suppress the conical magnetisation in NiCo₂O₄ is lower than that observed in pyrochlore lattice magnets.

We acknowledge the necessity of addressing this point and have added the following sentences.

Page 6, Right Column

We added the following sentence and corresponding references:

'The easy-cone magnetic anisotropy and DM interaction manifest the magnetic structural change from ferromagnetic to MTQ conical magnetic structure on pyrochlore lattice in the antisite NiCo₂O₄ thin film.

Since the magnetic interactions are weaker than the exchange interactions, QuadAHE is suppressed at lower magnetic fields compared to the AHE in frustrated magnetic pyrochlore materials.'

[Comment 2]

(2) In the manuscript, the QuadAHE is triggered by field-cooling process under a low magnetic field of ± 0.1 Tesla. The transverse resistivity was measured under the field up to 4 Tesla (see Fig. 3a and Supplementary Fig. 2). All spins at the pyrochlore sites might be aligned along the field direction at such a high field. What is the effect of the applied 0.1 Tesla during the field-cooling process under this situation? Did authors consider the effect of magnetic domains, which may play a dominant role in magnetization and AHE at low magnetic fields. Is it possible that the orientated magnetic domains[3] or phase separation[4] are enhanced during the process of field-cooling at ± 0.1 Tesla, then cause the anisotropic transverse resistivity at low temperature? The Hall results in the literature show the similar anomaly at low field. [4]

[3] Joonghoe Dho, Jungbae Kim, *Thin Solid Films* 756, 139361 (2022).

[4] M. Xue et al., *ACS Appl. Electron. Mater.* 2, 3964–3970 (2020).

[Response 2]

Thank you for pointing this out. We agree with this comment regarding the domain structure of MTQ. The conical magnetic structure of NiCo₂O₄ arises from the competition between exchange interaction and magnetic anisotropy with second-order term at least (may also be Dzyaloshinskii-Moriya interactions). The change from non-collinear to collinear magnetic structure is not a discontinuous change like a first-order phase transition but a continuous change like a crossover. In fact, the present results suggest that, even at high magnetic fields, the effects of anisotropy and Dzyaloshinskii-Moriya interaction remain, and the magnetic order is not in a fully collinear state. Therefore, it is assumed that the vestige of MTQ sign is not lost when the highest magnetic field in this measurement, 9 T, is applied and that the MTQ single domain structure is realised even when returned to a lower magnetic field.

The occurrence of QuadAHE remains consistent at cooling magnetic fields of 9 T and 0.1 T. Moreover, its antisymmetric behaviour with respect to H_z and H_{FC} dismisses the possibility that the QuadAHE results from oriented magnetic domains [3] or phase separation [4].

We acknowledge the necessity of addressing this point and have added the following sentences.

Page 2, Right column

We have added the following sentences:

'However, it cannot explain the antisymmetric behaviour to the reversal H_z and H_{FC} . Hence, the possibility of AHE from an extrinsic origin, such as the orientated magnetic domains [3] or phase separation [4], is also excluded in the present sample'.

Page 6, Right column

We added the following sentence:

'However, the experimental results confirm the existence of the same QuadAHE curve even after returning from a higher magnetic field of 9 T to a lower magnetic field (<0.3 T). Upon applying a magnetic field, the MTQ conical magnetic structure gradually changes with decreasing cone angle; however, it could not become completely collinear due to the influence of the remaining easy-cone magnetic anisotropy, preserving stable MTQ sign information'.

[Comment 3]

(3) According to $\rho_{xy} = R_0 H + S_A \rho_{xx}^n M$, magnetization generally have influence on the transverse resistivity. Consequently, the anisotropic M vs. H curves with both field-cooling and zero-field-cooling processes should be presented.

[Response 3]

As the reviewer pointed out, the transverse resistance ρ_{xy} has a component proportional to the out-of-plane magnetisation M_z and the in-plane longitudinal resistance ρ_{xx} . The MTQ conical magnetic structure causes anisotropy in the in-plane longitudinal resistivity but is sufficiently small compared to the isotropic component. Hence, the conventional transverse resistance ρ_{xy} is observed as a nearly isotropic signal. In the present experiment, σ_{xy}^{OI} is the signal corresponding to the M vs H curve. In fact, it is consistent with the M-H curve measured by the vibrating sample magnetometer (VSM). The agreement between σ_{xy}^{OI} and σ_{ZFC} suggests that field cooling and zero-field cooling do not produce different M vs H curves. In addition, other magnetisation measurements, such as the VSM, only observe M_z , so signals reflecting in-plane anisotropy cannot be detected.

We acknowledge the necessity of addressing this point and have added the following sentence.

Page 2, right column

'... the curve is identical to σ_{xy}^{OI} , indicating that the cooling process does not affect the Mz-H curve, ...'

[Comment 4]

(4) The behavior of field dependence of even-parity transverse resistivity (see Fig. 2 of Supplemental Material) is similar to that of magnetoresistance reported in previous literature [5]. Could the even-parity resistivity be caused by the magnetoresistance signal of Hall electrodes? Did authors consider the effect of Hall electrodes misalignment and the inhomogeneity of films?

[5] X. Xu et al., J. Appl. Phys. 132, 020901 (2022).

[Response 4]

We thank the reviewer for pointing out the important issues about artefacts from Hall electrodes. We cannot completely rule out the possibility that the symmetric (even-parity) transverse resistivity is manifested due to the Hall electrode misalignment or the film inhomogeneity. However, we believe that such components are minor and that the symmetric transverse resistivity resulting from changes in magnetic structure is dominant. In actuality, the symmetric transverse resistivity is almost reversed by rotating the cooling magnetic field's tilting direction (see Supplementary Fig. 6), suggesting that it is predominantly derived from the intrinsic origin of the magnetic structure.

To clarify this point, we have added the symmetric TR result to Supplementary Fig. 6 and a related statement as follows.

Page 4, in the Supplementary,

We have added the following sentence:

'In general, it is impossible to totally remove experimental errors, such as misalignment of Hall bar and film inhomogeneity, from the symmetric TR $\rho_{E_{ij}}^{\text{TR}}(H)$. However, it appears only below the temperature of T_S , concurrent with the magnetic transformation, as shown in Fig. 3d and Supplementary Fig. 4. Moreover, the sign shows the antisymmetric response with respect to H_{FC} , as shown in Supplementary Fig. 6, suggesting that the influence of these artefacts on the symmetric TR is negligibly small'.

Supplementary Fig. 6

Data for field cool dependence of symmetric TR were also added.

[Comment 5]

(5) In the manuscript, the authors claimed a magnetically induced electron liquid crystal state and cited the Ref. 17. Whether the electronic liquid crystal phase in NiCo₂O₄ and the new phase in Sr₃Ru₂O₇ (Ref. 17) have the same formation mechanism? Is the electron liquid state in NiCo₂O₄ the same as nematic phase observed in other strongly correlated electron systems, for instance high-temperature superconductors.

[Response 5]

Firstly, due to feedback from other reviewers, we have modified the term 'the electron liquid crystal state' to 'the electron nematic state'.

We thank the reviewer for the question about the origin of the electron nematic state. The liquid crystal-like electron nematic phase with broken C₄ symmetry has been discussed in various strongly correlated electron systems, such as the pseudogap phase of high-T_c cuprate superconductors, the hidden-order phase of the heavy-fermion superconductor URu₂Si₂, and the quantum critical point region of Sr₃Ru₂O₇. Strong Coulomb repulsion between electrons, superconductivity, and multiple orbital degrees of freedom

have been discussed in those compounds as the origin of the novel electron-ordered phases. In contrast, since the electron nematic state in NiCo₂O₄ is a completely spin order-originated deformed electronic state, we estimate that it stems from a different mechanism from that in Sr₃Ru₂O₇. We are convinced that our results open new developments in the electron nematic states and will greatly advance our understanding of conduction phenomena in extended multipoles.

To clarify this point, we have added a statement as follows.

Page 7, Left column,

We added the following sentence:

'The nematic electronic state arising from the distorted electronic state that stems from extended magnetic multipoles is presumed to exhibit a different origin than the electronic states observed in Sr₃Ru₂O₇ and other systems'.

[Comment 6]

(6) Authors showed the temperature dependence of odd-parity Hall effect (ρ_{OA}) in Fig. 3d. What is the angular dependence of ρ_{OA} on the angle between current and [100] axis? Could authors plot a similar figure as the Fig. 3 in Ref. 18 [6]?

[6] J. Wu et al., Nature 547, 432 (2017).

[Response 6]

We thank the reviewer for their valuable suggestion regarding the measurement setup. The angular dependence of ρ_{OA} on the angle ϕ between the current and [100] axis should be proportional to $\sin(2\phi)$, as shown in Eq. (2). To demonstrate the angular dependence of the current, measurements using the 'sunbeam' lithography pattern, as depicted in Figure 2 of J. Wu et al., Nature 547, 432 (2017), would yield more comprehensive results that support our transport model. However, we could not promptly prepare such electrodes and perform such an experiment due to current limitations in our experimental resources. We are confident that the experimental data provided in the manuscript are adequate for discussing transport symmetry, and we kindly request your acceptance of these data for now. We will consider the reviewer's suggestion as a potential future experimental plan.

Reviewer #2 (Remarks to the Author):

In this paper, Koizumi et al. present a comprehensive study of anomalous Hall effect in epitaxial NiCo₂O₄ films deposited on (001) MgAl₂O₄ substrates. The authors observe a topological-Hall-like feature in AHE switching hysteresis in field-cooled samples, which is absent in zero field cooling. This unconventional Hall signal shows a quadrupole dependence on the in-plane current direction and flips sign upon reversing the direction of the cooling field. Based on symmetry arguments, the authors attribute the anisotropic Hall effect to the emergence of an electron liquid crystal (ELC) state originating from an extended toroidal quadrupole (MTQ) conical magnetic order.

Overall, the observed anisotropic AHE in NCO is very interesting, but it is not an intrinsic property of NCO rather due to antisite disorder of Ni³⁺, making it less controllable. The proposed scenario of emergent ELC and MTQ order does not have solid experimental support and is far-fetched. The data may be explained by far more trivial transport model. As a result, I cannot recommend the publication of the paper in Nature Communications in the present form. Below are some detailed comments.

[Response]

We thank the reviewer for evaluating our manuscript positively and giving constructive suggestions, which helped improve the present manuscript.

Despite the reviewer's expressed interest in our results, we consider that publication of our results in Nature Communications was not recommended due to the reviewer's concerns that the results may have originated from the planar Hall effect. We also initially considered the possibility of the planar Hall effect but could not explain the present results totally by the planar Hall effect, as discussed below. Through careful analyses, we concluded that the effect is derived from MTQ conical structure based on Onsager's reciprocal theorem and symmetry considerations. The model is the most plausible from our results and consistent with almost all experimental results. In addition, the MTQ conical magnetic structures can also be theoretically elucidated to be stabilised in epitaxially distorted spinel ferromagnets due to the easy-cone anisotropy and the Dzyaloshinskii-Moriya interaction. Therefore, we believe that our conclusions are not 'far-fetched'. The QuadAHE is a novel Hall effect, distinct from the planar Hall effect, and we believe that our results meet the special criteria of Nature Communications.

As pointed out, the magnetic structure of MTQ is not directly determined in the current study and is less controllable due to its antisite origin. Since it is a thin-film sample, it is very difficult to analyse the magnetic structure by neutron scattering, which requires the volume of the sample. Moreover, due to the wide variety of sample synthesis and substrate parameters, the conditions under which MTQ becomes stable have not yet been found, and it will take much time to elucidate them. Additional research is required to gather more

evidence that will further support our model; nevertheless, we have confidence in the reliability of the samples and reproducibility of experimental data, and thus we believe that the current results are satisfactory to validate the model.

Here is a point-by-point response to the reviewers' comments and concerns.

[Comment 1]

1. The paper lacks some basic sample information. Please provide the film thickness of the two samples studied, surface morphology (e.g., AFM image and surface roughness), device details, e.g., whether it is measured via Hall bar or van der Pauw geometry, the type of metal electrodes, device fabrication/lithographic process, etc.

[Response 1]

We thank the reviewer for the suggestion. We acknowledge the lack of sample information and have added the corresponding sentences as follows.

Page 1, Right Column

We added the following sentence to explain the details of thin film samples:

'The anisotropic Hall effects were demonstrated on epitaxial NiCo₂O₄(001) films on MgAl₂O₄(001) substrates. NiCo₂O₄ is a conductive inverse spinel oxide exhibiting ferrimagnetism with a Neel temperature as high as $T_N=400$ K. Stoichiometric thin films exhibit perpendicular magnetic anisotropy at room temperature. Moreover, sharp out-of-plane magnetisation switchings are observed in the magnetic field dependence of AHE over the entire temperature range. In contrast, the antisite Ni³⁺ distribution, manipulated by the O₂ flow rate during thin film deposition, changes the magnetic anisotropy from the perpendicular to an easy-cone magnetic anisotropy at low temperatures. In the present study, we found that such antisite NiCo₂O₄ thin film realises a conical ferrimagnetic structure comprising out-of-plane ferromagnetic and in-plane MTQ components.'

Page 3, Figure 2 caption

We added the following sentence:

The measurements setup, e.g. Hall bar shape and current direction, is illustrated in the inset schematic of each figure.

Page 2, Line 9 and 12 of Supplemental Material

We added the following sentence.

'The film thickness, estimated from the X-ray reflectivity, is 50 nm. The a- and c-axis lattice constants are 8.08 Å and 8.20 Å, respectively'.

[Comment 2]

2. Please note that AHE in NCO has been reported in previously studies (e.g., Chen et al., Adv. Mater. 31, 1805260 (2019); Kan et al., PRB 104, 134407 (2021); Chen et al., APL 120, 242401 (2022)). Please cite the relevant work properly and discuss the current work in the context of existing literature.

[Response 2]

Thank you for pointing this out. We acknowledge that our current manuscript lacks sufficient references to previous studies. We have attempted to cite relevant studies as appropriately as possible and have added the following text, comparing our research with previous studies.

Page 1, Right Column

We added the following sentence and suggested references:

'The anisotropic Hall effects are magnetic and in-plane MTQ components'.

Page 4, Right column

'Such a magnetic structure from collinear to multistep non-collinear has not been observed in stoichiometric samples [Chen et al., Adv. Mater. 31, 1805260 (2019); Kan et al., PRB 104, 134407 (2021); Chen et al., APL 120, 242401 (2022)]. The results suggest that the easy-cone magnetic anisotropy due to the antisite Ni³⁺ distribution in the present sample is essential for structural changes'.

Supplemental Material, Page 3, Line 4-6

We have added the following sentences in the revised supplemental material and included the needed references:

'In the present thin film, σ_{AHE} monotonically decreases below $T_S=130$ K due to the magnetic structure changes from the N'eel- to the Yafet-Kittel-type state. However, Kan et al. reported that σ_{AHE} is almost constant at low temperatures, whereas Chen et al. reported that the sign of σ_{AHE} can be reversed by lowering the temperature in thinner NiCo₂O₄ films. Previous reports have been of stoichiometric NiCo₂O₄ thin film samples exhibiting perpendicular magnetic anisotropy, whereas the present sample is an off-stoichiometric film. Therefore, the difference in σ_{AHE} is assumed to be correlated with the ratio of antisite Ni³⁺ distribution.'

[Comment 3]

3. The authors attribute the anisotropic AHE to the transitions from PMA to two types of non-collinear spin structures. As discussed in Ref. [26], the appearance of easy cone magnetic anisotropy is not intrinsic to NCO but originates from the antisite distribution of Ni³⁺ occupying the Td site. For example, the authors use the squareness of the AHE hysteresis to gauge the onset of spin canting and ECMA. However, for

high quality NCO films on MAO, the square-shape hysteresis is sustained down to the lowest temperature (e.g., 2 K in Chen et al., Adv. Mater. 31, 1805260 (2019); Chen et al., APL 120, 242401 (2022)) in a wide range of film thickness. The fact that the QuadAHE varies from sample to sample (mentioned in Supplementary) further attests to the extrinsic nature of the effect. Such discussion is absent in the introduction/discussion of the manuscript. Please elaborate on this point in the main text so that the readers are properly informed.

[Response 3]

We agree that the magnetic structure remains collinear with the perpendicular magnetic anisotropy even at the lowest temperature in high-quality stoichiometric NiCo_2O_4 samples. As noted by the reviewer, the distribution of Ni^{3+} antisites at the Td site contributes to the stabilisation of easy-cone magnetic anisotropy. The amount of antisite can be controlled by adjusting the oxygen partial pressure during thin film synthesis [Shen et al., Physical Review B 101, 094412 (2020)]. Moreover, an increase in Ni^{3+} results in NiCo_2O_4 thin films exhibiting easy-cone magnetic anisotropy [Koizumi et al., Physical Review B 104, 014422 (2021)]. Therefore, the stability of the easy-cone magnetic anisotropy is influenced by extrinsic factors. Furthermore, another parameter appears to stabilise the MTQ conical magnetic structure, even in the presence of easy-cone magnetic anisotropy. Although the controlling parameter has not been unveiled yet, we have confirmed the expression of QuadAHE in all the samples we have synthesised and indicated the easy-cone magnetic anisotropy. This finding leads us to believe that the magnetic structure originates from an extrinsic factor; however, once the MTQ conical magnetic structure is realised, the QuadAHE exhibits an intrinsic nature. Since this discussion was absent in the original manuscript, we have included the following text in the introduction part in the revised manuscript.

Page 1, Right Column

We added the following sentence and those references:

'The anisotropic Hall effects are magnetic and in-plane MTQ components'.

Page 4, right column,

We added the following sentence and those references:

'Such a magnetic structure from collinear to multistep non-collinear has not been observed in stoichiometric samples [Chen et al., Adv. Mater. 31, 1805260 (2019); Kan et al., PRB 104, 134407 (2021); Chen et al., APL 120, 242401 (2022)]. The results suggest that the easy-cone magnetic anisotropy due to the antisite Ni^{3+} distribution in the present sample is essential for structural changes'.

[Comment 4]

4. In Fig 1a-d, the combination of 3D device schematic and 2D J-E relation is very confusing. Please better separate these two schematics and add corresponding laboratory coordinates.

[Response 4]

We thank the reviewer for the suggestions about the diagram in Fig 1a-d. We have revised Fig. 1 in line with the reviewer's suggestion. To enhance readability, we have excluded irrelevant figures from the main body of the paper or relocated them to the supplementary material.

[Comment 5]

5. Please define angle ϕ in Eq. (2) and add it to the schematic in Fig. 2.

[Response 5]

We thank the reviewer for pointing this out. ϕ is the angle between the direction of the current and the [100] axis. We have added the text describing the definition of angles and included the corresponding angles in Fig. 2.

[Comment 6]

6. The main problem of the paper is there is no substantial experimental evidence of the emergent ELC and MTQ order other than the angular dependence of AHE. The angular dependence in Eq. 2 can be simply due to the planar Hall effect (Li et al., J Vac Sci Technol A 40, 010807 (2022)), which can appear in the presence of in-plane magnetization (due to spin-canting, not toroidal magnetic order). It naturally explains the sign change of Hall signal upon reversing the cooling field, as switching the out-of-plane magnetization can reverse the spin canting direction. The planar Hall signal is also suppressed at high perpendicular field as the in-plane magnetization is quenched. This means the relevant physics may be the in-plane magnetocrystalline anisotropy and spin-orbit interaction.

[Response 6]

We express our gratitude to the reviewers for highlighting crucial issues in this paper. Upon initial examination of the results, we also considered the possibility of a planar Hall effect (PHE) resulting from in-plane magnetisation. PHE is a symmetric phenomenon with respect to the magnetisation (M), and the sign of PHE remains unchanged even when the in-plane magnetisation is rotated by 180° . This characteristic fails to explain the QuadAHE, which exhibits antisymmetric behaviour with respect to the external magnetic field. To achieve PHE sign inversion, a magnetic structural change, such as a 90° rotation of the in-plane magnetisation component, is necessary. Nevertheless, such a response cannot reasonably occur by reversing the out-of-plane field or cooling the magnetic field. Consequently, we confidently assert that the observed phenomena in this study distinctly differ from the PHE resistivity.

Conversely, the symmetric transverse resistivity originating from the Yafet-Kittel-type coplanar magnetic structure can be interpreted as the PHE resistivity arising from the in-plane antiferromagnetic component. Based on the reviewer's suggestion, we discuss the relationship with PHE, including explaining the differences with QuadAHE as described above. The following text has been included in the revised manuscript.

P. 1, Right Column,

We added the following sentence.

'From Onsager's theorem, the Hall effect which is the antisymmetric TR for H_z and M_z is isotropic with respect to the current direction, while a symmetric TR can be anisotropic. Indeed, a planar Hall effect, where TRs are anisotropic depending on the angle between J and M , shows a symmetric response to M . However, here, we demonstrate a manifestation of an unconventional anisotropic Hall effect even for charge current in a conical ferrimagnet composed of MTQ, as shown in Fig. 1c'.

P. 2, Right Column,

We added the following sentence.

'At first glance, such an anisotropic result appears to involve a planar Hall effect caused by assuming the presence of in-plane magnetisation. However, it cannot explain the antisymmetric behaviour with respect to H_z and H_{FC} . The possibility of AHE from an extrinsic origin, such as the orientated magnetic domains [Dho, et al. *Thin Solid Films* **756**, 139361 (2022)] or phase separation[Xue et al., *ACS Appl. Electron. Mater.* **2**, 3964 (2020)], is also excluded in the present sample for the same reason. Hereafter, the anisotropic Hall effect will be referred to as a quadrupole AHE (QuadAHE) to distinguish it from the conventional AHE and the planar Hall effect'.

P. 4, Left Column,

We added the following sentence.

'The symmetric and anisotropic TR ρ_{xy} can be regarded as the planar Hall effect originating from the in-plane antiferromagnetic component of the coplanar Yafet-Kittel-type magnetic structure'.

Reviewer #3 (Remarks to the Author):

[Comment]

The authors have reported the anisotropic Hall effect by systematically analyzing transverse resistivities with respect to applied current directions. They have also discussed the origin of anisotropy based on cluster (extended) multipoles which characterize magnetic structures according to the crystallographic symmetry. The experiment and its theoretical analyses appear sound, but it is quite unclear whether their findings are really significant in this and related fields. The main reason of unclearness is quite low readability of the manuscript that contains many inaccurate jargons, and undefined words. Therefore, I do not recommend this work as it is published in Nature Communications. I give several points below that seem to enhance low readability with quite inaccurate use of technical terms, and they make the essential statement of this work obscure. The presentation of the manuscript gives an impression that the work is not well organized and summarized.

[Response]

We appreciate the reviewer for the evaluation of our experimental results and theoretical analysis. As the reviewer pointed out, we regret that some terminologies were not correct, well-defined, or readable. We have been able to incorporate changes to reflect most of the suggestions provided by the reviewers. We have highlighted the changes within the manuscript.

Below, is the point-by-point response to the reviewers' comments and concerns.

[Comment 1-1]

(1) In the abstract, the authors use "topological Hall effect" for systems with any of nontrivial magnetic structures. However, this is quite crude and a topological Hall effect has its own meaning and definition.

[Response 1-1]

As the reviewer pointed out, the explanation of the 'topological Hall effect' was unclear. We have revised the explanation as follows.

In abstract:

'... arises in nontrivial magnetic structures such as skyrmion and chiral magnets.' was revised as '... describes the appearance of a transverse resistivity with applied charge current in ferromagnets or non-collinear magnets with nonzero scalar spin chiralities.'

[Comment 1-2]

Similarly, "electron liquid crystal state" is not well established, and the present system is just a deformed band structure from conical magnetic ordering, and has nothing to do with "liquid crystal".

[Response 1-2]

Thank you for pointing this out. We agree with the reviewer and have modified the term 'electron liquid crystal state' to 'an electron nematic state'. We recognised that the electron nematic state is a more appropriate term, as it indicates a state of matter in which electronic correlations spontaneously break the rotational symmetry of the underlying lattice [H. Yamase, V. Oganesyan, and W. Metzner, Phys. Rev. B 72, 035114 (2005)].

We have changed the related terminologies throughout the revised manuscript and added some explanations and related references.

[Comment 2]

(2) The Onsager reciprocal relations are the statement about the off-diagonal components of the resistivity in this context as the authors exhibit in the sentence above Eq.(1), and it does not assume the antisymmetric property with respect to H and M as Eq.(1). Therefore, it is not surprising that the in-plane anisotropy of resistivity tensor appears provided the symmetric components appear in the tensor, which gives rise to anisotropic responses in total.

[Response 2]

We appreciate the reviewer's insightful remarks regarding the nature of our results. As rightly pointed out, Onsager reciprocal relation does not assume the presence of only antisymmetric components with respect to H and M. If symmetric components are included in the off-diagonal terms, it is plausible to expect the manifestation of anisotropic transverse resistance in total. However, in this study, we performed a symmetrisation analysis to separate the symmetric and antisymmetric components with respect to H and M. Although the anisotropic behaviour for the symmetric component is a trivial result, that for the antisymmetric component is a nontrivial result for Onsager's reciprocal relation. Hence, we believe that the present results are 'remarkable'.

Onsager reciprocal relation for the anisotropic response is assumed to have a symmetric property with respect to H and M and the time reversal operation. The most remarkable finding in this study is a parameter generating transverse resistivity derived from magnetic structure, which is symmetric with time reversal operation but antisymmetric with respect to H and M. This property yields nontrivial anisotropic and antisymmetric transverse resistivity with respect to H and M. Such a Hall effect mechanism has not been previously discovered; therefore, we believe that these unprecedented findings satisfy the requirements of Nature Communications.

We acknowledge the necessity of addressing this point and have made the necessary modifications in the revised manuscript.

[Comment 3]

(3) The authors use the terms, even-parity and odd-parity, for Hall effect, which are quite unclear. Moreover, the word "Hall effect" is usually used to refer to the antisymmetric components in the resistivity tensor, since the symmetric components can be eliminated by appropriate principal-axis transformation. The main message of this manuscript seems to be the existence of both symmetric and antisymmetric components due to the conical spin structure accompanying (cluster) magnetic dipole and magnetic toroidal quadrupoles, however, it may be smeared out by inaccurate use of words.

[Response 3]

Thank you for pointing this out. We agree with this comment. It was inappropriate to use the term, even-parity and odd-parity, for the Hall effect. We also agree that the Hall effect should refer only to the antisymmetric component in the transverse resistivity with respect to H and M. Therefore, we have modified these terms to 'symmetric transverse resistivity (TR)' and 'Hall effect', respectively, throughout the manuscript.

[Comment 4]

(4) The condition of disappearance of anomalous Hall effect is not the same as no (zero) Berry curvature, since the former is determined by the sum of the latter over the occupied electronic states. However, there are several inaccurate statements in the manuscript.

[Response 4]

We thank the reviewer for pointing out the important mistake. As the reviewer suggested, our paper contained an incorrect statement. We have corrected the relevant statements in the revised manuscripts.

[Comment 5]

(5) The authors use "Mz-even" and "Mz-odd" and so on, in which the use of even/odd is quite unclear.

[Response 5]

Thank you for bringing to our attention the use of unclear terms. We acknowledge that these words can be confusing and have made the necessary modifications. We have replaced 'Mz-even/odd' with 'symmetric/antisymmetric with respect to Mz'.

[Comment 6]

(6) In my opinion, the authors quite frequently use their own words without definitions, and they are not familiar to the readers in the related fields. I should strongly recommend to improve sentences and English expressions.

[Response 6]

We appreciate the reviewer for feedback regarding the readability of the manuscript. Accordingly, we have made revisions throughout the text to minimise uncommon abbreviations and provide clear definitions of technical terms to enhance readability.

REVIEWER COMMENTS

Reviewer #1 (Remarks to the Author):

Prof. Yanagihara and colleagues made great effort to respond to the comments raised by reviewers. After reading all of the responses, I still found that authors did not provide sufficient experimental evidence to support the conclusions in the revised manuscript.

1. M versus H data need to be shown for understanding the evolution of magnetic domains with temperature and magnetic field, as well as comparison with electrical transport results.

2. The effect of 0.1 Tesla field during the FC process is still unclear. What happens if a magnetic field of 0.5 Tesla or 1 Tesla is applied? Would we get transverse resistivity with different behavior?

During the measurement, the applied field is much higher than 0.1 Tesla, does the field-cooling process with low field really play an important role?

3. The transverse resistivity data presented in the manuscript were calculated by authors. However, they cannot rule out the contribution of extrinsic factors, such as misalignment of electrodes and film inhomogeneity.

4. The authors proposed an electron nematic state in NiCo₂O₄, but did not provide any direct evidence.

Owing to the above reasons, I cannot recommend this manuscript to be published in Nature Communications.

By the way, in Figure 3d, the label “Smymmetric TR” is incorrect and should be “Symmetric TR”.

Reviewer #2 (Remarks to the Author):

In the revised version, the technical quality and clarity of the paper are much improved. The authors have adequately addressed many, but not all, of my previous comments. I can recommend the publication of this work in Nature Communications if the following comments can be properly addressed.

1. Previous Comment 1: The authors provided sample thickness but did not include the characterization of sample morphology and device fabrication details. Please add AFM topography measurements and give the surface roughness. Also, how are the devices fabricated. For example, what are the dimensions of the active areas of the Hall bars? From the Supplementary Fig. 6, there are multiple Hall bar devices fabricated on the same substrate. How are different devices/electrodes isolated from each other? Does it involve etching or depositing on per-patterned substrates?

2. Previous Comment 4: Please add laboratory coordinates in Fig. 1 schematics to improve clarity, e.g., define x-y-z axes in the 3D schematics and x-y axes in the 2D schematics.

3. In the first sentence of the revised abstract, the authors stated that the AHE appears in "ferromagnets or non-collinear magnets with non-zero scalar spin chiralities". First, AHE appears where there is non-zero out-of-plane magnetization, so it can appear in ferrimagnets (as in the case of NCO) and sometimes antiferromagnets. Second, the description of non-collinear magnets with non-zero scalar spin chirality seems to incorporate the concept of topological Hall effect within the definition of AHE, which is an arguable attempt. Would it be clearer to describe them in the context of Berry curvatures in k-space or real space? Please consider revising this sentence for clarity, either separating these two scenarios or giving a more detailed description.

4. In the last paragraph on Page 6, the authors stated that "the magnetic interactions are weaker than the exchange interactions." Please give the corresponding energy scales and cite the relevant references.

5. In the Supplementary Note 1, the authors mentioned that the substrates for the two samples are from two providers and the samples exhibit different AHE. What is the reason for using different providers? Can the authors elaborate on the effect of substrates on the observed AHE?

6. As I mentioned in my previous review, my major concerns are that there is no direct characterization of the magnetic structure of the MTQ order and the antisite disorder is not controlled. I consider that the observed AHE effect is very interesting and worth publishing, and the proposed scenario is plausible. However, it is important to have these points elaborated in the manuscript so that the readers are informed about the limitations. I suggest including the related response (last paragraph on Page 7 of the rebuttal letter), from "the magnetic structure of MTQ is not directly determined" to "Additional research is required to gather more evidence that will further support our model", in the discussion of the main text.

Reviewer #3 (Remarks to the Author):

The authors replied all of my requests in the revised version, and I conclude that they are appropriate. Although the readability is still not satisfactory maybe due to the organization of presentation, and quite a few extraordinary expressions, it manages to convey the authors assertion. Thus, I recommend it for publication.

Reviewer #1 (Remarks to the Author):

[Comment]

Prof. Yanagihara and colleagues made a great effort to respond to the comments raised by reviewers. After reading all of the responses, I still found that authors did not provide sufficient experimental evidence to support the conclusions in the revised manuscript.

[Response]

The authors thank the reviewer for evaluation of our response to the comments and for suggestions to improve the manuscript. Below are our responses to all of the reviewer's comments.

[Comment 1]

1. M versus H data need to be shown for understanding the evolution of magnetic domains with temperature and magnetic field, as well as comparison with electrical transport results.

[Response 1]

The authors thank the reviewer for the important comment. Following the suggestion, we added such M versus H data, measured by the vibration sample magnetometer (VSM), at Supplementary Fig. 8a with comparing the anomalous Hall effect after zero field cooling (ZFC) [figure below]. The magnetization response to H is totally consistent with AHE, indicating the evolution of magnetic domains can be revealed via AHE with ZFC. The temperature dependences of M vs H curves are also shown in the previous paper [H. Koizumi, *et al.* Phys. Rev. B **104**, 014422 (2021)]. We have added descriptions of these points along with the added figure.

[Comment2]

2. The effect of 0.1 Tesla field during the FC process is still unclear. What happens if a magnetic field of 0.5 Tesla or 1 Tesla is applied? Would we get transverse resistivity with different behavior?

During the measurement, the applied field is much higher than 0.1 Tesla, does the field-cooling process with low field really play an important role?

[Response 2]

The authors thank the reviewer for pointing out the important issue. Cooling with the application of a slightly tilted magnetic field is an essential procedure to achieve a homogeneous domain of MTQ conical magnetic structure and to observe QuadAHE. The magnitude of the cooling magnetic field H_{FC} required to realize the single magnetic domain depends on the temperature at which cooling is to be started [see Supplementary Fig. 8]. For example, if cooling is started at 300 K, the AHE vs. H curves obtained for $\mu_0 H_{FC} = 0.1$ T and 9 T are consistent as shown in the figure below. However, it exhibits a different behavior when ZFC ($\mu_0 H_{FC} = 0$ T). Thus, even a cooling field as low as 0.1 T is important to homogenize the MTQ conical domain, and once cooled, the single domain state remains robust during AHE vs. H measurements with applied higher magnetic field up to 9 T. We added the H_{FC} dependence of AHE at Supplementary Fig. 8b and corresponding descriptions.

[Comment 3]

3. The transverse resistivity data presented in the manuscript were calculated by authors. However, they cannot rule out the contribution of extrinsic factors, such as misalignment of electrodes and film inhomogeneity.

[Response 3]

The authors thank the reviewer for the comment. Since the antisymmetrization analysis can extract only the components that are reversed by the magnetic field, the contribution of extrinsic factors independent with a magnetic field can be ruled out from the raw data of transverse resistivity. On the other hand, symmetric transverse resistivity may include extrinsic effects such as the electrode misalignment and inhomogeneity. However, the magnetic field-dependent component of the symmetric TR component disappears at high temperatures, as shown in supplementary Fig. 4d. Therefore, the component of transverse resistivity due to extrinsic factors can be evaluated to be sufficiently smaller than the transverse resistivity originating from the magnetic structure. We added descriptions of these points to the Supplementary Note. 4.

[Comment 4]

4. The authors proposed an electron nematic state in NiCo₂O₄ but did not provide any direct evidence.

[Response 4]

The authors thank the reviewer for the comment. For example, anisotropic Fermi-surface states observed by ARPES would provide more direct evidence for nematic electronic states. We have prepared to perform ARPES experiments on samples in which this anisotropic anomalous Hall effect is observed. However, the expected distortion of the Fermi surface is about 0.5%, which may be difficult to observe with the resolution of the equipment available to us.

On the other hand, transverse resistance is a technique that can detect Fermi surface distortions with higher sensitivity. There are several papers claiming that the anisotropy of the transverse resistivities are originating from nematic electronic states [2-4]. Therefore, it is reasonable to claim the existence of an anisotropic Fermi surface corresponding to the nematic electronic state from the observation results of this study.

We added a description about the limitation of current study and the lack of direct evidence of nematic state to the conclusion part.

[2] Rafael M. Fernandes et al., Phys. Rev. Lett. 107, 217002 (2011)

[3] Jonatan Wårdh et al., PNAS Nexus, 2, pgad255 (2023).

[4] Wu, J. et al. J Supercond Nov Magn 32, 1623–1628 (2019).

Reviewer #2 (Remarks to the Author):

[Comment]

In the revised version, the technical quality and clarity of the paper are much improved. The authors have adequately addressed many, but not all, of my previous comments. I can recommend the publication of this work in Nature Communications if the following comments can be properly addressed.

[Response]

The authors thank reviewer for evaluation of the revision. We regret that we could not fully responded to reviewer previous comments. We have added experimental data and descriptions and are able to respond to all comments this time.

[Comment 1]

1. Previous Comment 1: The authors provided sample thickness but did not include the characterization of sample morphology and device fabrication details.

[Response 1]

The authors thank reviewer for pointing out important issues about sample. Our response to all comments are shown in below.

[Comment 1-1]

Please add AFM topography measurements and give the surface roughness.

[Response 1-1]

We have performed AFM topography measurement in NiCo₂O₄ thin film. The results were added on Supplementary Fig. 7f [see also figure below]. A surface roughness and peak-to-valley distance estimated in measured regions are approximately 0.3 Å and 3.6 Å, respectively.

[Comment 1-2]

Also, how are the devices fabricated. For example, what are the dimensions of the active areas of the Hall bars?

[Respond 1-2]

We made two patterns of Hallbar of clover- and rectangle-shape, as shown in Fig. 2 and Supplementary Fig. 3, respectively. Those sizes are $20\ \mu\text{m} \times 300\ \mu\text{m}$ and $200\ \mu\text{m} \times 1400\ \mu\text{m}$. The information of Hall bar was added in the Method section.

[Comment 1-3]

From the Supplementary Fig. 6, there are multiple Hall bar devices fabricated on the same substrate. How are different devices/electrodes isolated from each other? Does it involve etching or depositing on pre-patterned substrates?

[Response 1-3]

The different devices/electrodes are isolated by the Ar ion milling technique. Since the substrate MgAl_2O_4 is an insulator, each hole bar is electrically separated by removing the NiCo_2O_4 film other than the hole bar by the milling. The fabrication method of Hallbar was added in the Method section.

[Comment 2]

2. Previous Comment 4: Please add laboratory coordinates in Fig. 1 schematics to improve clarity, e.g., define x-y-z axes in the 3D schematics and x-y axes in the 2D schematics.

[Response 2]

Following the reviewer's suggestion, the laboratory coordinates, x-y-z axes in the 3D schematics and x-y axes in the 2D schematics, were added in Fig.1.

[Comment 3]

3. In the first sentence of the revised abstract, the authors stated that the AHE appears in "ferromagnets or non-collinear magnets with non-zero scalar spin chiralities". First, AHE appears where there is non-zero out-of-plane magnetization, so it can appear in ferrimagnets (as in the case of NCO) and sometimes antiferromagnets. Second, the description of non-collinear magnets with non-zero scalar spin chirality seems to incorporate the concept of topological Hall effect within the definition of AHE, which is an arguable attempt. Would it be clearer to describe them in the context of Berry curvatures in k-space or real space? Please consider revising this sentence for clarity, either separating these two scenarios or giving a more detailed description.

[Response 3]

The authors thank the reviewer for pointing out the important issue. Following the reviewer's suggestion, we revised the sentence based on the context of Berry phase as follow.

In abstract:

'Berry phases in both momentum and real space cause transverse motion in itinerant electrons, manifesting various off-diagonal transport effect such anomalous and topological Hall effects.'

[Comment 4]

4. In the last paragraph on Page 6, the authors stated that "the magnetic interactions are weaker than the exchange interactions." Please give the corresponding energy scales and cite the relevant references.

[Response 4]

The authors thank the reviewer for pointing out this point. The sentence was ambiguous and misleading. For more clarity, we have revised it as follows and added references.

The last paragraph on Page 6:

'Since the magnetic interactions are weaker than the exchange interactions' was modified to 'Since the energy scale of magnetic anisotropy and DM interactions are weaker than the antiferromagnetic superexchange interactions.'

We added the following reference:

[5] T. Moriya, Phys. Rev. Lett. **4**, 228 (1960).

[Comment 5]

5. In the Supplementary Note 1, the authors mentioned that the substrates for the two samples are from two providers and the samples exhibit different AHE. What is the reason for using different providers? Can the authors elaborate on the effect of substrates on the observed AHE?

[Response 5]

The authors thank the reviewer for the comment. To check the influence of substrate quality on QuadAHE, we fabricated some samples using substrates obtained from different providers. Their AHE showed different AHE vs H curves, however the reproducibility of QuadAHE itself was confirmed. We revised and added related description in the Supplementary Note 1.

[Comment 6]

6. As I mentioned in my previous review, my major concerns are that there is no direct characterization of the magnetic structure of the MTQ order and the antisite disorder is not controlled. I consider that the observed AHE effect is very interesting and worth publishing, and the proposed scenario is plausible. However, it is important to have these points elaborated in the manuscript so that the readers are informed about the limitations. I suggest including the related response (last paragraph on Page 7 of the rebuttal letter), from “the magnetic structure of MTQ is not directly determined” to “Additional research is required to gather more evidence that will further support our model”, in the discussion of the main text.

[Response 6]

The authors thank the reviewer for the suggestion. We added descriptions about the limitations of present study and the future perspective in the conclusion part.

Reviewer #3 (Remarks to the Author):

[Comment]

The authors replied all of my requests in the revised version, and I conclude that they are appropriate. Although the readability is still not satisfactory maybe due to the organization of presentation, and quite a few extraordinary expressions, it manages to convey the authors assertion. Thus, I recommend it for publication.

[Response]

We thank the reviewer for evaluating our manuscript positively and recommending it for publication. In this revision, we are improving the readability as possible.

REVIEWERS' COMMENTS

Reviewer #1 (Remarks to the Author):

After two revisions, the authors answered all my questions, the manuscript has been improved a lot. I would like to recommend it to be published in Nature Communications.

Reviewer #2 (Remarks to the Author):

In the revised version, the authors have satisfactorily addressed all my comments. I can recommend the publication of the work in Nature Communications.